# Marine Macro-Litter (Plastic) Pollution of German and North African Marina and City-Port Sea Floors

Gerald Schernewski [1,2,*], Gabriela Escobar Sánchez [1,2], Philipp Wandersee [1,3], Xaver Lange [4], Mirco Haseler [1] and Abdallah Nassour [3]

[1] Coastal & Marine Management Group, Leibniz-Institute for Baltic Sea Research, Seestrasse 15, D-18119 Rostock, Germany; gabriela.escobar@io-warnemuende.de (G.E.S.); philipp.wandersee@t-online.de (P.W.); mirco.haseler@io-warnemuende.de (M.H.)

[2] Marine Research Institute, Klaipeda University, Universiteto Ave. 17, LT-92294 Klaipeda, Lithuania

[3] Department of Waste and Resource Management, Rostock University, Justus-von-Liebig-Weg 6, D-18059 Rostock, Germany; abdallah.nassour@uni-rostock.de

[4] Department of Physical Oceanography and Instrumentation, Leibniz-Institute for Baltic Sea Research, Seestrasse 15, D-18119 Rostock, Germany; xaver.lange@io-warnemuende.de

\* Correspondence: gerald.schernewski@io-warnemuende.de; Tel.: +49-381-519-7207

**Abstract:** The macro-litter (plastic) sea-bottom pollution of 14 city harbors and marinas in North Africa and in the western Baltic Sea was investigated using a new simple mobile underwater camera system. The study was complemented by a harbor-manager survey and 3D hydrodynamic transport simulations. The average pollution in German marinas was 0.1 particles/m$^2$ sea floor (0.04–1.75). The pollution in North African marinas on average was seven times higher (0.7 particles/m$^2$) and exceeded 3 particles/m$^2$ in city-center harbors. The resulting > 100,000 litter particles per harbor indicate the existence of a problem. At 73–74%, plastic particles are dominating. Existing legal and management frameworks explain the lack of plastic bottles and bags on sea floors in Germany and are one reason for the lower pollution levels. Items that indicate the role of untreated sewage water were not found. Harbor festivals seem to be quantitatively irrelevant for open sea-bottom pollution. Our method tends to underestimate the pollution level. Model simulations indicate that storms can cause litter reallocations and sediment cleanings. However, marina sea-floor monitoring is recommendable because it addresses pollution hotspots, is cost-effective and takes place close to emission sources. Further, the effectiveness of land-based pollution-reduction measures can easily be assessed.

**Keywords:** recreational harbor; harbor sediment; monitoring; Baltic Sea; Hanse Sail Rostock; Kiel Week; hydrodynamic model; waste water; indicator; policy

## 1. Introduction

It is estimated that 250,000 metric tons of plastic float in the seas and that macro- and meso-plastic (>5 mm) has a weight share of 86% in it [1]. Marine macro-plastics (>25 mm) can be distinguished into floating and sinking material, depending on their density. Bio-fouling with marine biota increases the density of plastics and within weeks turns floating into sinking plastic [2]. Floating material can be transported by wind and currents over long distances [3], but a large share is accumulated at beaches in the surroundings of emission pathways from surface-wave activity [4]. Sinking plastics can be accumulated on the sea bottom, especially in sheltered bays or in the deep sea. Canals et al. [5] provide a comprehensive overview of the present global state. However, large shares of plastics that accumulate on sandy coastal sea bottoms are subject to wave-induced resuspension and in the end largely end up at the coastline, as well [6]. The accumulation of amber at beaches visualizes this process [7]. Altogether, near the coast, beaches and sheltered coastal areas, such as bays and harbors/ports, are likely hotspots for plastic accumulation [8,9].

The emitted quantities of macro-plastics are correlated to human activities, population densities and modified by human behavior and the effectiveness of waste-management systems. Large rivers, coastal urban areas as well as untreated sewage and stormwater are considered to be major pathways for macro-plastic to the marine environment [10]. Additionally, intensively used touristic beaches can also serve as a source for macro-plastic in the marine environment and not only as a sink [10]. Human activities, especially in and around urban coastal areas, have to be regarded as of highest importance for marine plastic emissions [11].

In most parts of the Baltic Sea, coastal city centers do not host industrial ports any more, but the city seasides have been transformed into leisure and recreation areas, hosting leisure-boat marinas and offering water access for locals and tourists. Further, during the last decades, the Baltic Sea became an inner European Union sea. This development fueled water-related tourism, and the number of sport-boat ports increased. Additionally, many city harbors developed additional infrastructures to host the increasing number of cruise ships and ferry lines as well as the associated tourism. In 2017, over 5 million passengers visited the Baltic ports onboard cruise ships. The number of calls was about 2500 cruise ships and is still increasing [12]. Additionally, in 2023, 86 regular ferry routes existed in the Baltic Sea [13], and in 2020 about 67 million passengers embarked and disembarked from the over 200 larger commercial Baltic Sea ports [14]. Around the Baltic Sea about 95 coastal cities are located as well as many smaller seaside resorts, and the vast majority have a city port and host marinas. This number underlines the increasing potential importance of city- and sport-boat ports as plastic-emission and pollution hotspots in the Baltic Sea region, which is similar to other regions [15].

In the recent assessment of the state of the Baltic Sea, marinas and leisure harbors are considered as a pressure [16], and marine litter is taken into account as an indicator. However, reliable data on litter (plastic) sea-floor pollution in harbors are scarce [17]. From a fishing port in Turkey, over 300 kg of seabed litter was extracted in 2017 [18]. In 2020, in Newport harbor (USA) scuba divers collected over 2000 kg of marine litter [19]. In 2022, in Mülheim harbor (Germany) three divers collected 200 kg of marine litter within 2 h [20]. From the city port of Rostock (Germany), 550 kg of marine litter was removed during one day in 2022 [21]. These data indicate that a largely unknown pollution problem in harbors seems to exist.

In Alexandria, Egypt, Shabaka et al. [22] describe high microplastic concentrations at beaches and floating near the water surface in harbors. In Tunisia, about 1 kg of waste is produced per inhabitant per day, and it is assumed that a tourist generates more than twice as much [23]. In all North African countries, high plastic and litter emissions into coastal and marine systems results from waste mismanagement [24]. It can be expected that intensive tourism, together with waste mismanagement and local fisheries, causes a strong litter pollution of harbor sea-floors in North Africa.

Kühn et al. [25] provide a review of the consequences of marine litter in the marine environment and on wildlife, such as entanglement and ingestion. Another problem is the toxicity of chemicals associated especially with plastic litter [26]. Furthermore, marine litter is known to serve as a habitat and dispersal vector [27]. Last but not least, litter is a nuisance that negatively affects public perception and can have negative consequences for tourism [28].

The question is, why is there no marine-litter-pollution monitoring in harbors if harbors are pollution hotspots? Presently, the marine litter monitoring in the European Union (EU) follows the requirements of the Marine Strategy Framework Directive (MSFD) [29], which aims at a good environmental status for the European regional seas and coasts. As a consequence, the monitoring focusses on open seas and remote beaches and not on pollution hotspots.

Potentially, several methods exist that are suitable for marine litter (plastic) monitoring in harbors. Dredging and trawling are mechanical options. Optical methods use videos and images collected by divers, using high-resolution cameras installed on remotely operated

vehicles, or using towed cameras. Acoustic methods use, for example, side-scan sonar, synthetic aperture sonar or multibeam echosounder systems. Madricardo et al. [30] and Hanke et al. [31] provide a comprehensive overview of applied methods. The problems of existing approaches are that they are relatively high cost and effort, as well as that they are often unsuitable for marinas with a complex shape and large number of boats [32]. Therefore, a new city port and marina-pollution-screening method is needed that is simple and cost-effective. It should focus on providing data that enables pollution-reducing measures and allows for the assessment of their effectiveness.

Our overall objective is to provide an insight into harbor sea-floor pollution with macro-litter (plastic) in contrasting areas in the Baltic Sea region and in North Africa, by combining field studies and monitoring with literature reviews and modeling. This includes the development and application of a cost-effective harbor-bottom plastic-screening method.

## 2. Study Sites and Methods

### 2.1. Study Area

The sea-bottom-pollution study was carried out in a total of 14 marinas in North Africa and in the western Baltic Sea. In North Africa, sampling took place in two marinas in the city of Alexandria in Egypt (6.1 million inhabitants) and in three marinas in Tunisia, namely the city of Bizerte (about 143,000 inhabitants), Yasmine Hammamet (a tourism resort and part of the city of Hammamet, the latter has about 100,000 inhabitants) and Monastir (about 93,000 inhabitants). The criteria for selecting the harbors were their accessibility and their location in an urbanized area. All African marinas have touristic infrastructure and beaches in the vicinity (Figure 1).

In the Baltic Sea, the studies focused on harbors surrounding the cities of Rostock and Kiel. Rostock (210,000 inhabitants) and Kiel (246,000 inhabitants) are located in the western Baltic Sea and are the most important Baltic Sea harbor cities in Germany. Both host cruise, cargo, dry and military ports as well as shipyards. In addition, each city has 10 marinas for leisure boats. Sampling took place in four of the marinas in Rostock (Alter Strom, Hohe Düne, Alter Stadthafen and Ludewigbecken) and in five marinas in and around Kiel (Düsternbrook, Seeburg, Strande, Wendtorf and Laboe) (Figure 2).

With the Kiel Week and the Hanse Sail Rostock, the cities host major annual Baltic sailing and harbor events. The Kiel Week is the largest sailing event in the world and the largest summer festival in the Baltic Sea region. In 2023, it attracted 3.8 million visitors over nine days, the highest number ever recorded. More than 100 larger historic steam and sailing ships offered boat trips, over 4000 sport-sailors with about 1500 boats took part in the competitions and 22 cruise ships visited the harbor during the event [33]. The Hanse Sail was founded in 1991. In 2023, the Hanse Sail Rostock counted about 500,000 visitors during four days and about 150 ships offered boat trips for the visitors [34]. Before the COVID-19 pandemic, for example in 2009, the Hanse-Sails counted more than 1 million visitors and in some years more than 200 participating historic ships [35]. The potential role of these events on sea bottom pollution was one aspect of interest.

### 2.2. Screening Method for Macro-Litter on the Bottom of Marinas

For the sea-bottom screening a fast mobile system was developed, tested and improved in several iteration cycles. An underwater camera and an underwater light were installed on a 3 m carbon telescope-stick. The camera was connected to a smartphone via a coax cable. The smartphone App GoPro Quick, GoPro, Inc., San Mateo, CA, USA was used to control the underwater camera and to transfer, store and view the photos and videos on the smartphone above the water. During each sampling start- and end-time, cloud cover, wind speed [m/s], estimated wave height [m], water depth [m], water transparency/Secchi depth [m] camera depth and spatial coverage of the underwater photos were protocolled. The system was applied from bridges, catwalks, quay walls and piers. Depending on the structure and size of the marina, between 2 and more than 20 different spots in a marina were investigated, and the surveyed area ranged from 7–8 m$^2$ in Montazah and

Ludewigsbecken to 756 m$^2$ in Hohe Düne, Rostock. A total of 1938 m$^2$ of harbor bottom was sampled in Germany and 596 m$^2$ in North Africa. Data collection took place between March and July 2022 [36].

Before the sea-floor pictures were taken, the suitable height of the camera above the sea bottom was determined. The acceptable distances between camera and sea floor were between 0.25 m and 1.6 m and depended on water transparency (Secchi depth) and water depth. On average, the distance was about 40% of the Secchi depth and about 35% of the water depth. The present system was suitable for a maximum water depth up to 4 m, which was sufficient for all harbors.

The underwater photos were analyzed visually, with bare-eye observation of the photos on a large computer monitor, according to Watters et al. [37]. The identified litter items were documented and categorized according to the OSPAR (Convention for the Protection of the Marine Environment of the North-East Atlantic) guidelines for monitoring marine litter on beaches [38]. A video-based analysis was tested but not applied for field-data collection.

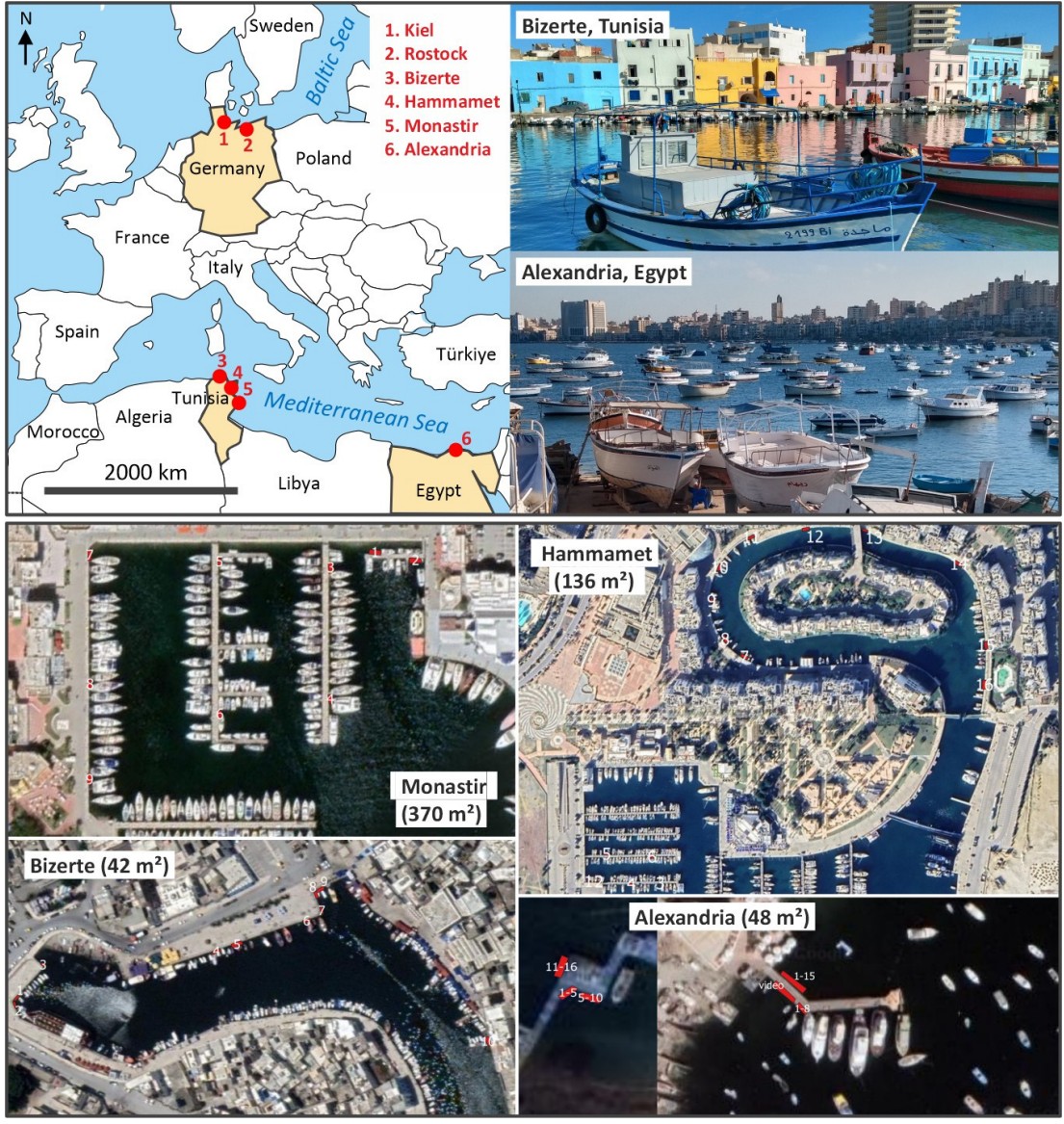

**Figure 1.** General overview of the sample sites in Tunisia, Egypt and Germany as well as all North African harbors including name, investigated sea-bottom area (m$^2$) and concrete sampling locations within the harbors (numbered red areas). Satellite images by GoogleMaps.

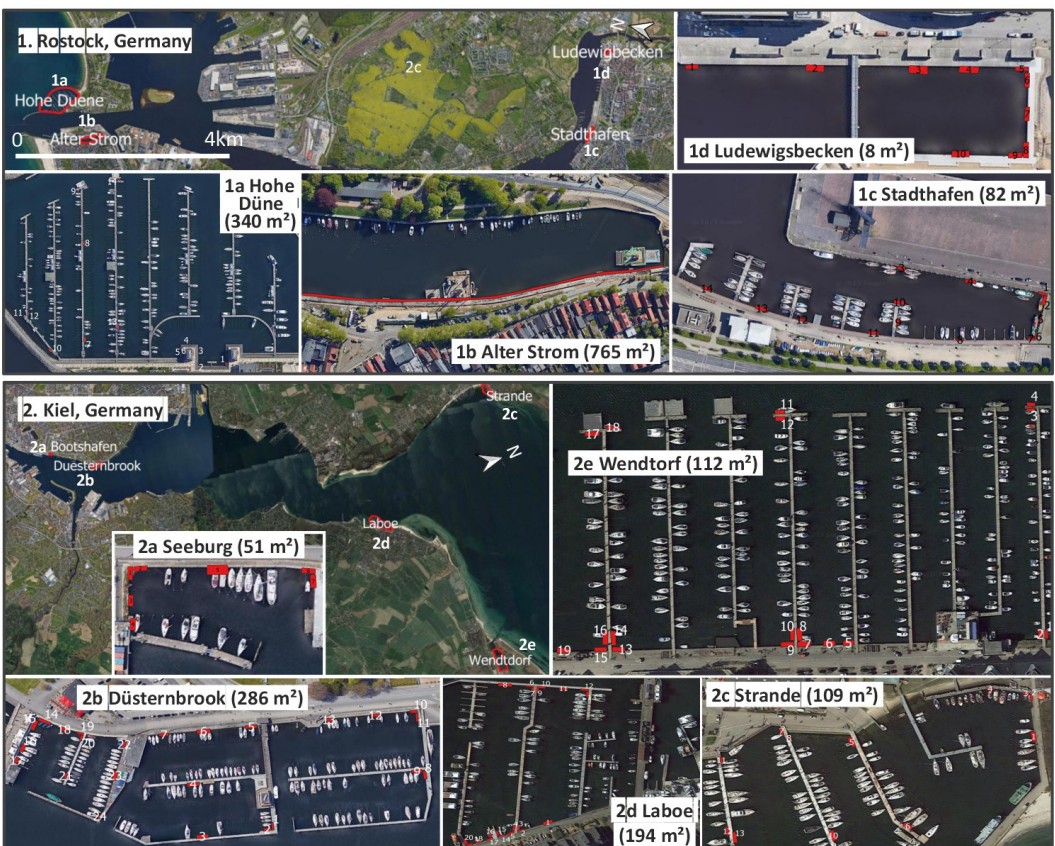

**Figure 2.** Details of the harbors in Rostock and Kiel, Germany, including name, investigated sea-bottom area (m²) and concrete sampling locations within the harbors (numbered red areas). Satellite images by GoogleMaps.

The total costs for equipment were 666 EUR, including the underwater camera (GoPro Hero 8, GoPro, Inc., San Mateo, CA, USA, 225 EUR) and the smartphone (Huawei P8 Lite, Shenzhen, Guangdong, China,167 EUR), an additional underwater light (60 EUR) and the telescope stick (60 EUR). The time required to cover one square meter of harbor bottom ranged from 0.3 to 10 min (median about 1 min), depending on the degree of pollution and water transparency (Figure 3). The total time for analyzing the photos was about 50 h or 1 min/m² sea floor.

*2.3. Model Approach*

Exemplary model simulations were carried out for Rostock harbor covering the entire Warnow Estuary including the coastal parts of the western Baltic Sea. The model simulations covered two Hanse Sail events, because the Hanse Sail is most likely the largest emission event of macro-litter into the Rostock city harbor. The aim was to gain insight into whether sinking litter is deposited locally at the sea bottom near the emission site or is transported over longer distances.

The simulations involved two steps: first, a 3D hydrodynamic model calculated hindcast simulations for the years 2009 and 2010, and second, the results were used for an offline Lagrangian particle-tracking approach. For the former, the General Estuarine Transport Model (GETM) [39,40] with the General Ocean Turbulence Model (GOTM) [41] as the turbulence closure model was used. The study area is numerically discretized on a structured grid with a horizontal resolution of 20 m and a vertical resolution of 25 terrain-following sigma layers. The meteorological forcing is calculated from the output of the reanalysis product of the German Weather Service (COSMO-REA6) with a temporal resolution of one hour and a spatial resolution of 6 km. Discharge data of the river

Warnow were included as daily mean values. Lange et al. [42] provide details on boundary conditions and model validation.

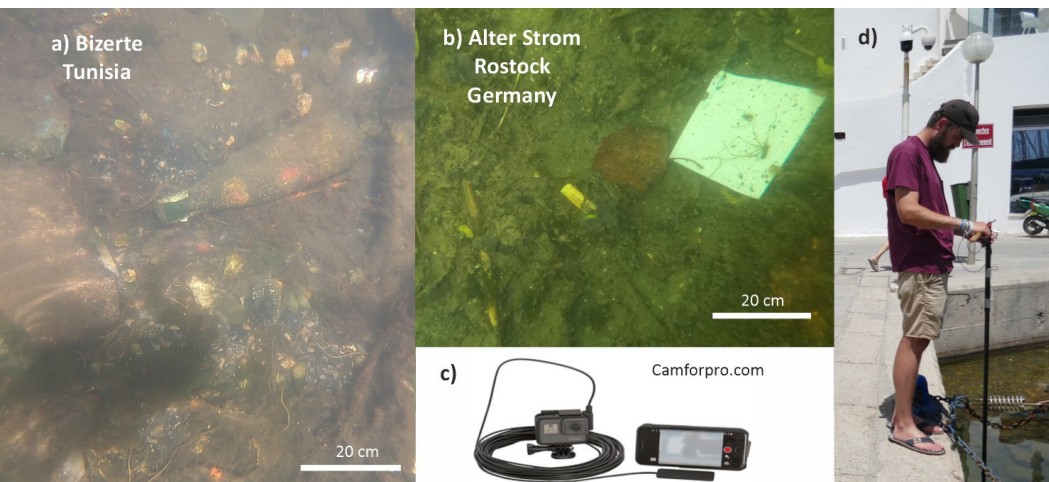

**Figure 3.** Exemplary underwater photos from a Tunisian (**a**) (Mediterranean Sea) and a German (**b**) (Baltic Sea) marina as well as the technical setup (**c**) and the method for field application (**d**).

The main output parameters were staggered horizontal current velocities on an Arakawa C-grid, horizontal eddy viscosity calculated using a Smagorinsky parameterization, and bottom shear-stress based on a quadratic drag, stored with 5 min resolution each. These were used as forcing inputs for the particle-tracking model, Ocean Parcels [43]. Diffusion was considered using the Milstein scheme (1st order), and a critical stress condition for near-bottom particles was implemented (see below). Particles were allowed to beach in coastal sections characterized by reeds.

### 2.4. Scenario Simulations

The model simulations addressed only sinking litter (plastics). The most relevant sinking plastic polymers are rigid polyvinyl chloride (PVC, density 1.3–1.45 $g/cm^3$) and polyethylene terephthalate (PET, density 1.38 $g/cm^3$).

In the shallow city harbor, with a water depth of about 2.5–3.5 m near the harbor wall and up to 6 m in the shipping channel, common sinking plastic particles such as PVC or PET settle to the harbor bottom within minutes. At the bottom, the particles were assumed to be transported with the 50 cm layer of bottom water. Particles deposited on the sediment require a higher bottom shear stress to be remobilized and transported. We assumed that only the highest 25% of the bottom shear stresses, calculated during the simulation period, would cause a transport. These optimistic assumptions generally favor a near-bottom transport.

The model and Lagrangian plastic-transport simulations covered the Hanse Sail years of 2009 and 2010. The background is that Hanse Sails are likely causing pollution and that several events were accompanied by intensive data collection and field activities on litter emissions. The years 2009 and 2010 were chosen because they had contrasting weather conditions and were among the years with the highest number of visitors. The particle-tracking simulations always started at the beginning of the Hanse Sail and ended after 10 days. Additionally, one simulation assumed that the strongest storm of the year 2009 (strong gale; maximum wind speed of 21.9 m/s) occurred immediately after the Hanse Sail. The aim was to gain insight into the potential reallocation of bottom litter under extreme conditions.

The field data of several Hanse Sail years allowed an estimation of floating-litter emissions, but were too weak to provide data on sinking-litter emissions into the estuary. For the model simulations, we assumed (expert guess) an emission of litter (plastic) with a density above 1 $g/cm^3$ of 0.3 emitted macro-litter particles per 1000 visitors per day.

The scenario hypothetically assumed the emission of one sinking plastic particle per 300 Hanse Sail harbor coastline (about 2.1 km), emitted hourly during the Hanse Sail opening hours (Thu. to Sun. between 12:00–24:00 o'clock). That means, 91 particles were emitted per day and 364 particles were emitted altogether during the whole event.

## 3. Results

### 3.1. Litter Pollution in Harbors—Concentrations

The marinas in Rostock differ in location and framework conditions. The Alter Strom in Rostock, Warnemünde, is a former fishing harbor in the center of the seaside resort of Warnemünde. The Alter Stadthafen is the oldest harbor basin in Rostock city center, and the Ludewigsbecken is an old but recently restored harbor area. In principle, all harbors have already existed for over 500 years and are nowadays used for leisure boats. All of them are affected by major harbor events, namely the Warnemünde Week and the Hanse Sail. While the Alter Stadthafen and the Alter Strom show a pollution of 0.12–0.14 litter particles/m$^2$, the Ludewigsbecken shows a higher pollution (1.75 particles/m$^2$). The Baltic Sea marina in Hohe Düne, opened in 2005, shows a lower pollution of 0.02 particles/m$^2$ (Table 1).

**Table 1.** Results of the marina-bottom marine-litter screening. Analyzed area (%) indicates the share of the harbor area that was covered with our method. Secchi depth indicates the water transparency. Camera distance is the distance between camera lens and sea floor.

| | Marina | Analyzed Area (m$^2$) | Harbor Area (m$^2$) | Analyzed Area (%) | Number of Items | Items per m$^2$ | Items per Harbor | Water Depth (m) | Secchi Depth (m) | Camera Distance (m) |
|---|---|---|---|---|---|---|---|---|---|---|
| **Rostock** | Alter Strom | 765 | 38,000 | 2.01 | 105 | 0.14 | 5,216 | 2.40 | 2.40 | 0.78 |
| | Alter Stadthafen | 82 | 10,000 | 0.82 | 10 | 0.12 | 1,220 | 3.10 | 0.80 | 0.63 |
| | Ludewigsbecken | 8 | 11,000 | 0.07 | 14 | 1.75 | 19,250 | 1.40 | 0.60 | 0.25 |
| | Hohe Düne | 340 | 250,000 | 0.14 | 6 | 0.02 | 4,412 | 3.90 | 3.30 | 1.43 |
| **Kiel** | Düsterbrook | 286 | 30,000 | 0.95 | 21 | 0.07 | 2,203 | 2.40 | 2.40 | 0.95 |
| | Seeburg | 51 | 3,000 | 1.70 | 9 | 0.18 | 529 | 1.30 | 1.30 | 0.70 |
| | Laboe | 194 | 110,000 | 0.18 | 7 | 0.04 | 3,969 | 2.20 | 2.20 | 0.86 |
| | Strande | 109 | 30,000 | 0.36 | 6 | 0.06 | 1,651 | 2.50 | 2.50 | 0.80 |
| | Wendtorf | 112 | 65,000 | 0.17 | 13 | 0.12 | 7,545 | 1.90 | 1.90 | 0.70 |
| **Egypt** | Montazah | 7 | 190,000 | 0.00 | 21 | 3.00 | 570,000 | 1.60 | 1.75 | 0.39 |
| | Yachtclub | 41 | 700,000 | 0.01 | 23 | 0.56 | 392,683 | 1.75 | 1.75 | 0.83 |
| **Tunisia** | Bizerte | 42 | 40,000 | 0.11 | 156 | 3.71 | 148,571 | 1.75 | 1.40 | 0.54 |
| | Yas. Hammamet | 136 | 160,000 | 0.09 | 59 | 0.43 | 69,412 | 2.70 | 2.70 | 0.81 |
| | Monastir | 370 | 75,000 | 0.49 | 168 | 0.45 | 34,054 | 3.80 | 3.80 | 1.60 |
| | **Germany** | **1947** | **547000** | **0.71** | **191** | **0.28** | | | | |
| | **North Africa** | **596** | **1165000** | **0.14** | **427** | **1.63** | | | | |

Similar to Rostock, the city harbor in Kiel has existed for over 500 years. The traditions of the leisure-boat harbors Düsterbrook and Seeburg, near the city center, go back to the Olympic Games of 1936. Similar to the city harbors of Rostock, both show a pollution of 0.07–0.18 litter particles/m$^2$. Both harbors are principally affected by the Kiel Week. In contrast, the Baltic Sea marinas in the surroundings of Kiel show a pollution of less than 0.1 litter particles/m$^2$. Overall, the average pollution in German marinas is 0.28 particles/m$^2$ resp. 0.1 particles/m$^2$, if the total investigated area is divided by the total number of particles.

In Egypt and Tunisia, the harbors are old and still partly host traditional fishing activities and are located close to the city centers. The only exception is Yasmine Hammamet, which represents a new sport-boat harbor built in the late 1990s as a tourism center. The pollution in Montazah (Alexandria, Egypt) and Bizerte (Tunesia) exceeds 3 particles/m$^2$ (Table 1). The average pollution in African marinas is about 1.6 particles/m$^2$ resp. 0.7 particles/m$^2$, if the total investigated area is divided by the total number of items. The pollution is about seven times higher than in German harbors. The extrapolation of the data to the entire harbor area visualizes the dimension of the pollution problem. In the large African harbors, the total number of particles is above 100,000. However, these extrapolations are associated with very high uncertainties because, apart from the Alter Strom in Germany, less than 1% of the total harbor areas were investigated.

### 3.2. Litter Pollution in Harbors—Items

In German city harbors, cigarette butts have the highest share of all item classes with 32%. This indicates the dominant role of social activities in the harbors (Table 2). In contrast, in sport-boat marinas, ropes and textiles dominate with 42%. The latter represents classical items related to leisure boats. Plastic items account for 74% of all items.

**Table 2.** Item distribution in the marina-bottom marine-litter screening.

| Location | Plastic | | | | | Metal Can | Glass Bottle | Wood, Paper etc. | Total |
|---|---|---|---|---|---|---|---|---|---|
| | Cigarette Butt | Undefined Item | Rope & Textiles | Bag | Bottle | | | | |
| **Germany** | | | | | | | | | |
| City harbors | 36 | 18 | 24 | 5 | 0 | 3 | 5 | 22 | 113 |
| Sport boat marinas | 2 | 5 | 9 | 0 | 0 | 0 | 1 | 4 | 21 |
| **Egypt** | | | | | | | | | |
| City harbors/marina | 0 | 11 | 3 | 4 | 1 | 0 | 0 | 7 | 26 |
| **Tunisia** | | | | | | | | | |
| City harbors/marina | 48 | 23 | 69 | 17 | 20 | 21 | 8 | 35 | 241 |

The low absolute numbers of particles in harbors in Egypt do not provide a reliable insight into the item distribution. In harbors in Tunisia, cigarette butts (20%) also play an important role. In general, plastic bags and bottles have a higher share in African harbors than in Germany. Similar to Germany, plastic items account for 73% of all items. This means that in all harbors, plastic is the dominating material.

### 3.3. Factors Controlling Pollution—Harbor-Bottom Cleaning

In general, the observed pollution level and spatial differences are influenced and/or controlled by the emission level, possible harbor-bottom cleanings, resuspension and reallocation of particles to other areas by wave action and bottom currents, and, last but not least, our data-collection methods.

A survey and interviews with harbor managers were carried out to get an insight into harbor waste-management and the potential role of harbor-bottom cleanings. Only four surveys, including additional interviews, were obtained. In Strande, Germany, and Marina Cap Monastir, Tunisia, floating litter is removed with nets; in Yasmine Hammamet, Tunisia, floating litter is removed manually; and Schilksee, Germany, uses a Seabin. All four harbors have appropriate waste-collection systems in place, and floating litter appears to be removed as needed.

Two interviewees mentioned direct rainwater-outlets discharging into the harbor. Kiel and Rostock have combined sewer systems [44]. Wastewater and stormwater are treated jointly in wastewater-treatment plants and discharged into different parts of the city harbors. Heavy rains can temporarily cause sewer overflows and the discharge of untreated water. For other harbors, such as Bizerte, it is known that wastewater can enter the harbor temporarily. This means that wastewater-and-stormwater discharge can potentially play a role in harbor-bottom pollution. Hygienic items, such as cotton bud sticks, can be indicators of untreated sewage water. However, a detailed analysis of all items for each harbor does not indicate an important role of untreated sewage on sea-bottom pollution.

None of the interviewees were aware of any harbor-bottom cleaning. It seems that harbor-bottom cleaning is rare and most likely does not influence our data collection and the observed sea-bottom pollution.

### 3.4. Factors Controlling Pollution—Litter Resuspension and Reallocation

The observed pollution levels potentially might be affected by resuspension and reallocation of pieces by wave-induced turbulence and currents. This question can be addressed exemplary with 3D hydrodyamic model simulations. Further, the simulations provide an insight into the spatio-temporal behavior of sinking litter, especially plastic.

Figure 4 shows the transport of hypothetical litter emitted to the sea during the Hanse Sails in August. As said before, it can be assumed that the Hanse Sail is a major pollution event. The Hanse Sail years 2009 and 2010 show contrasting wind directions and wind speeds temporally exceeding 7 m/s (4 Beaufort; moderate breeze). The wind velocities in 2009 and 2010 were above the average that is usually observed during Hanse Sails. Therefore, it can be expected that the transport was more intensive compared to common Hanse Sail years.

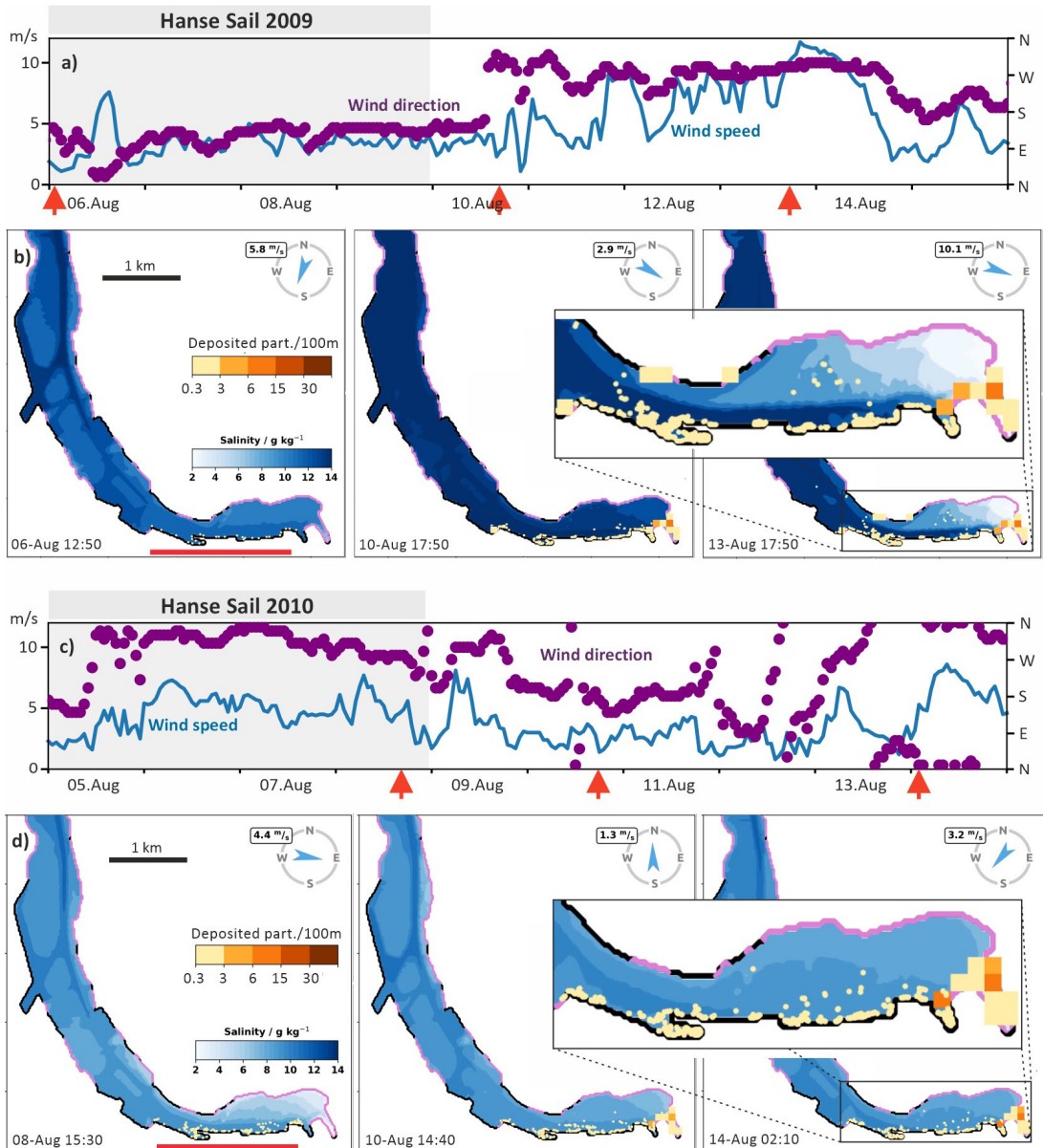

**Figure 4.** (**a**) Wind speed and direction during Hanse Sail 2009 which lasted from 6th to 9th August. The red arrows indicate the date and time of figure series (**b**). (**b**) Shows the model simulation of sinking litter and the litter deposition at the bottom or along the shoreline for three dates. It is assumed that altogether 300 particles were emitted during the event. (**c**,**d**) Show the same for the Hanse Sail 2010. Videos of the model simulations are in the Supplementary Material (S1).

The model simulations for Hanse Sails 2009 and 2010, do not indicate any significant transport of particles deposited on the sediment. Some of the emitted particles are transported along the shoreline and remain in the city-harbor area. Under normal summerly wind conditions, resuspension and reallocation of particles at the bottom does not play a

relevant role. Since the sinking particles largely remain very close to the emission spot, a local pollution is likely to occur, especially during events, such as the Hanse Sail. Especially in very sheltered areas located in the center of the event, such as the Alter Stadthafen, (Figure 2), it seems that a long-term accumulation of litter above the sediment can take place. However, the observed litter concentration at the bottom of the Alter Stadthafen was, at 0.12 particles/m$^2$, relatively low.

The marinas Seeburg and Düsterbrook in Kiel (Figure 2) are located close to the Kiel Week. These are marinas where a strong sediment pollution with litter is to be expected due to the Kiel Weeks. However, here too, the observed concentrations of litter on the sea floor were relatively low, with 0.18 resp. 0.07 particles/m$^2$.

The bottom transport is a complex interaction between external forcing from the Baltic Sea, estuarine circulation, harbor morphometry and wind-shelter effects, and can hardly be predicted without a model. Therefore, a model simulation was carried out to gain insight as to whether heavy storms have an effect on bottom-litter reallocation.

Figure 5 shows how the most severe storm observed in the year 2009, with wind speeds above 20 m/s from westerly directions, would have affected bottom-litter transport. The model suggests a resuspension of most litter and a transport towards up to 4 km north-west. During the storm, the model suggests a strong reallocation of bottom litter and an accumulation in the shipping channel and at the coastline. During the Hanse Sails 2009 and 2010, the average bottom-current velocity hardly ever exceeded 0.07 m/s. During the storm, the model calculated an average bottom velocity up to 0.12 m/s at times when the wind speed was close to or above 20 m/s. A reallocation of litter at the sea floor can happen, and it is likely that this affects our data and our observed litter concentrations.

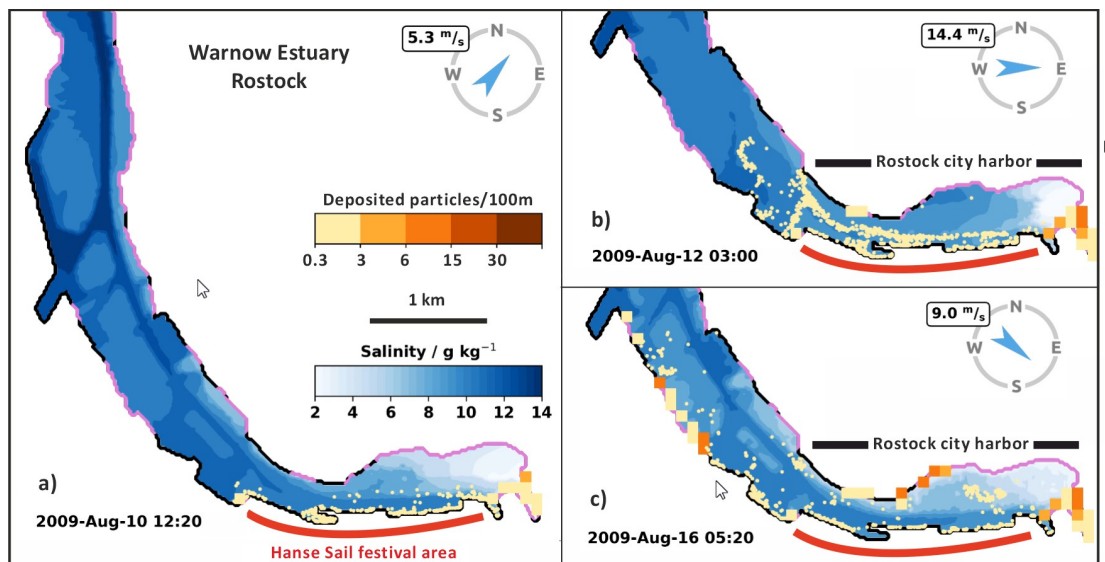

**Figure 5.** Emission of litter with a density above 1 g/cm$^3$ during the Hanse Sail festival and bottom-litter transport in the Warnow Estuary. It is assumed that altogether 300 particles were emitted during the event. (**a**) Transport of emitted litter in the Rostock city harbor after Hanse Sail (6–9 August 2009). (**b**,**c**) Litter resuspension and transport during a hypothetical storm after the Hanse Sail. The video of the model simulation is in the Supplementary Material (S1).

According to the model results, in the Warnow Estuary, it is very unlikely that any sinking litter emitted near the city center will reach the mouth of the Baltic Sea, at a distance of 11 km. The same is probably true for the Kiel Week and the Kiel Fjord because the framework conditions are comparable. Pollution of the open Baltic Sea bottom resulting from events in the city harbors is very unlikely. However, many of the 150 ships, participating in the Hanse Sail Rostock, transport visitors several times a day between the city harbor and the Baltic Sea. The emissions from boats are not included in the simulations.

## 4. Discussion

### 4.1. Evaluation of the Methodology

Our sea-bottom screening method has several advantages: The total cost for equipment was, 666 EUR, relatively low. With 0.3 to 10 min (median around 1 min), the time required to cover one square meter of harbor bottom is relatively short. The system is simple, easy to learn and easy to apply. It is mobile and portable, and allows relatively fast comparative studies in very different environmental and regional settings. Several minor technical improvements could further improve the method, such as a stronger underwater light, an increased stick length, or an automatic distance measurement between camera and bottom.

However, the method also has several weaknesses: Objects are often overgrown or covered with organic material. Therefore, a manual analysis of the photos with bare eyes is recommendable. Going into detail, this is time-consuming and the identification of objects is biased because it depends on the size, color, state and material of the object. Consequently, not all objects have the same likelihood of being found, and larger, colorful and freshly deposited items are favored. In addition, the method and the quality of the resulting data are highly dependent on the water transparency [32]. It is more suitable for clear-water seas, such as the Mediterranean Sea, and is problematic and time-consuming in turbid systems, such as coastal lagoons. How well and how long an object can be identified after deposition, depends on how fast it is overgrown or buried by sediment. This depends on environmental conditions, such as light conditions, nutrient availability and productivity, as well as sediment reallocation. These factors vary greatly from harbor to harbor. Therefore, for how long an identified object was located on sediment can hardly be determined. The method, and how we applied it, does allow the calculation of deposition rates, e.g., the number of pieces per $m^2$ and month. Despite a considerable time and effort, the number of particles found in several harbors was too low to allow a detailed item analysis.

The method is fast because it is carried out from solid harbor walls, wooden bridges or fixed floating pontoons. The resulting weakness is that only the nearby bottom pollution is assessed which can hardly be considered spatially representative. We have tried to compensate for this effect by using a large number of sampling locations within each harbor. However, in most cases less than 1% of the total harbor area was sampled.

For a spatially representative sampling, a boat or a float could be used, but this would increase the time required. Further, it would increase the difficulty of obtaining permission for field studies in harbors and marinas. Permissions were a major challenge in all African harbors and restricted the choice of harbors and the extent of our field studies. This was a major reason, too, for not applying underwater drones for our study. We tested the use of different underwater drones in German harbors but faced additional problems that made the approach ineffective [32].

### 4.2. Harbor-Pollution Monitoring and Model Simulations

Madricardo et al. [30] and Canals et al. [5] provide a comprehensive overview of sea-bottom-monitoring methods, and Hanke et al. [31] define detailed requirements for monitoring that addresses trends in the amount of litter deposited on the seafloor as part of the European Marine Strategy Framework Directive. Presently, the official monitoring focusses on shallow coastal waters, marginal seas and the deep-sea floor. Our approach, focusing on sea floors in urban areas, addresses complementary areas and pollution hotspots.

For shallow waters, underwater visual surveys by scuba divers are recommended and detailed protocols exist [31]. Presently, the scuba-diver method delivers the most reliable data for harbors and marinas, as well. Recently, it has been applied to detect lost fishing gear on harbor sediments in the Baltic Sea. This method is associated with high efforts and costs, which do not allow its integration into regular monitoring programs.

Usually, monitoring is intended to provide an insight into the pollution process, meaning how much litter is deposited per area and time-unit. A combination of scuba divers and our method could meet this demand. Scuba divers could be used to clean harbor floor areas from litter. Afterwards, the ongoing pollution process on these areas could

be observed with our method, and a combination with dredging methods could also be beneficial. However, our approach is certainly suitable as a screening method to get a first insight into pollution levels and major items. The results allow the definition of pollution reduction measures and can serve for assessing the effectiveness of these measures.

For the model simulations, we used an advanced spatially high-resolved 3D flow model that is very well adapted to the Warnow Estuary [35]. In contrast, the assumption that a litter resuspension and bottom transportation takes place at a bottom shear stress of $8.25 \times 10^{-7}$ Pa is a strong simplification. This does not take into account important factors, such as the different shape, size or density of litter particles. Therefore, the model simulations are only a first step towards a better understanding of bottom transport and have to be treated with caution. If we assume that what the model suggests is close to reality, we see that under common wind conditions, the resuspension and transport of litter above the sediment does not play an important role. During storms, a resuspension and reallocation of litter can take place even in sheltered harbor areas (e.g., Alter Stadthafen). Therefore, the observed litter concentrations above the sediment are likely to be affected by storms and represent an uncertain pollution time period. The low observed bottom-litter concentrations in city harbors suggest, that events, such as the Hanse Sail or the Kiel Week, are not important pollution events. This can be misleading because during storms litter reallocation and sediment cleaning may have taken place. All harbor-monitoring methods suffer from this uncertainty. The lack of comparable data and literature do not allow us to critically reflect on our results in more detail.

*4.3. Harbor Pollution—State, Causes and Management*

All harbors in and around Kiel and Rostock have sea-floor-litter concentrations below 0.2 particles/m$^2$. Marinas in semi-urban surroundings of the cities show concentrations below 0.1 particles/m$^2$. The only exception is the relatively small Ludewigsbecken in Rostock, which shows higher pollution (1.75 particles/m$^2$). It is likely that this higher concentration is still the result of recent reconstruction work. Centuries of human activities in and around the city harbors of Rostock and Kiel, such as frequent social activities as well as large-scale harbor events such as the Hanse Sail and the Kiel Week, with millions of visitors annually, do not seem to significantly increase the observed harbor-floor litter pollution. In all these harbors, bottom-cleaning activities are unknown and certainly do not take place regularly. Our monitoring method certainly underestimates the present pollution, as only recently deposited items have been identified.

In 2019, Tunisia counted 9.4 million tourist arrivals [45]. Tourism focusses on several coastal resorts and the season is between June and September. The three Tunisian harbors of Yasmine Hammamet, Monastir and Bizerte, are tourism hotspots throughout the summer season. These intensive human activities are reflected in the observed sea-floor pollution which is about seven times higher than German harbors. In city harbors, such as Bizerte and Montazah, the concentration exceeds 3 particles/m$^2$ sea floor. In Tunisian touristic harbors, measures to reduce emissions are implemented, but especially in cities, deficits in waste management and an insufficient public awareness of litter problems still seems to exist. Our concentrations observed in marinas are in the range of 100 to far above 1000 times higher than those of macro-litter studies in Baltic and Mediterranean coastal sea-floors [10,46,47]. An overview by Vlachogianni et al. [48] shows areas, such as the Gulf of Aqaba in the Red Sea, with bottom-litter concentration up to 2.8 pieces/m$^2$. However, our results indicate that harbor-bottom pollution is a problem especially in the African city harbors.

In the cities of Kiel and Rostock in Germany an improved waste-management system has been implemented in recent years. This especially affected the harbor events (Hanse Sail, Kiel Week). It includes a deposit system for cups, the mandatory use of degradable tableware, free waste deposit containers for all ships, reusable fence fasteners, low-emission fireworks, nightly ground cleaning and an optimized waste-bin distribution and emptying system [49]. These activities are largely a result of the European Union's ban on single-use plastic items such as plastic plates, cutlery, straws, balloon sticks and cotton buds

which came into force in 2021 [50]. Further, the Kiel Week received the Platinum Level Certification of the Clean Regatta Program. The 20 criteria include the elimination of single-use items (e.g., elimination of single-use water bottles, plastic straws, bags, dinnerware, water-refill stations) and a responsible waste management (e.g., green team, proper waste-bin placement and paperless signage for event management). The Kiel Week aims at a sustainable events certification according to ISO 20121 [51] and Rostock is following a similar pathway. In Germany, already since 2003, a deposit for plastic bottles exists, and it was expanded in 2021. Before, only 1 L bottles were charged with a 0.25 EUR deposit, but today, a deposit of at least 0.15 EUR exists for the vast majority of all plastic bottles. From 2022, certain types of plastic bags became prohibited and customers have to pay for most thicker plastic bags. These laws may explain the lack of plastic bottles and bags on sea floors in Germany.

Our results suggest that the existing waste management systems and the legal frameworks are suitable to reduce emissions and reduce harbor-bottom litter pollution. An improved environmental awareness and recognition of the litter problem have certainly reduced the emissions, as well. However, this needs to be explored in more detail.

## 5. Conclusions

We provide the first comparable macro-litter pollution data for marinas and city-harbor sea floors. The data is restricted to identifiable items located at the sediment surface. In general, sea-floor monitoring in marinas and city harbors has advantages. It addresses pollution hotspots, is cost-effective and is close to the emission sources. Further, the effectiveness of land-based pollution-reduction measures have become visible at their early stages, and the monitoring results provide fast feedback for improved measures. A screening of the littering process on the harbor landside is hardly possible because of an extreme spatio-temporal variability and many controlling factors. Litter on the sea floor shows only a very limited temporal variability and is much less affected by disturbing factors (e.g., cleaning). In smaller, largely closed and wind-sheltered harbors, a reallocation of deposited litter from bottom currents may not be frequent, but in larger harbors storm-induced litter-removal processes have to be taken into account.

**Supplementary Materials:** The following supporting information can be downloaded at: https://doi.org/10.5281/zenodo.8433800 (accessed on 1 July 2023). S1: Videos showing all particle tracking simulations of near-bottom marine macro-litter transport in Rostock (Warnow Estuary).

**Author Contributions:** Conceptualization, G.S., G.E.S. and P.W; methodology, P.W., G.S. and G.E.S.; validation, G.S., G.E.S. and P.W; model simulations, X.L.; formal analysis, P.W. and G.S.; writing, G.S. and P.W.; visualization, G.S., P.W. and X.L.; review and corrections M.H. and A.N.; supervision, G.S. and G.E.S.; project administration, M.H. and A.N.; funding acquisition, M.H., A.N. and G.S. All authors have read and agreed to the published version of the manuscript.

**Funding:** This research was funded by the BMU/ZUG project TouMaLi (Beitrag der nachhaltigen Abfallwirtschaft im Tourismus zum Schutz der Meeresökosysteme), grant number 65MM0001. Minor funding was provided by the German Federal Ministry of Education and Research project Coastal Futures, grant number 03F0911B, as well as by Interreg South Baltic project COP, grant number STHB.02.03.-IP01-0006/23. GES also received support from the Doctorate scholarship program in Ecology and Environmental Sciences at Klaipeda University, Lithuania.

**Institutional Review Board Statement:** Not applicable.

**Data Availability Statement:** The data presented in this study are available on request from the corresponding author.

**Acknowledgments:** We would like to thank Miriam von Thenen and Esther Robbe for supporting the field campaigns in Africa and Margaux Gatel-Rebours for graphical support.

**Conflicts of Interest:** The authors declare no conflict of interest.

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
