# Peer review of "Marine Macro-Litter (Plastic) Pollution of German and North African Marina and City-Port Sea Floors"

_applsci, doi:10.3390/app132011424_

Round 1

Reviewer 1 Report

Sampling was done in select three countries: Germany, Tunisia and Egypt. The title can be narrowed down accordingly, instead of saying European and African Marina. A detailed survey on the migration of plastic particles of various size ranges in the Baltic Sea is reported, and a hotspot of plastic pollution is identified.

Is it not possible to identify the pollution hotspots by monitoring the social behaviour in the local cities under investigation? Since the hydrodynamics of the sea depends on several external factors, what is the reliability of this model for future prediction? Please emphasis the research importance of such an elaborate survey. 

Quality of English is good

Author Response

Thank you for your readiness to review our paper and for your constructive comments. Please find our response to your comments below:

Since you indicated that the description of the methodology can be improved, we added a paragraph:

“Before the sea floor pictures were taken, the suitable height of the camera above the sea bottom was determined. The acceptable distance between camera and sea floor were between 0.25 m and 1.6 m and depended on water transparency (Secchi depth) and water depth. In average the distance was about 40 % of the Secchi depth and about 35 % of the water depth. The present system was suitable for a maximum water depth up to 4 m, which was sufficient for all harbors.”

Additionally, Table 1 was extended and now shows complementary data, namely Secchi depth (water transparency), water depth and the distance between camera lens and sea floor.

R1 comment: The title can be narrowed down accordingly, instead of saying European and African Marina.

Response: The title has been changed into: “Marine macro-litter (plastic) pollution of German and North African marina and city port sea floors”

R1 comment: Is it not possible to identify the pollution hotspots by monitoring the social behaviour in the local cities under investigation?

Response: The work was carried out within a larger project on marine litter, called TouMaLi. Relating the behavior to the pollution level and searching for efficient measures to improve the situation is one task, that is now ongoing.

R1 comment: Since the hydrodynamics of the sea depends on several external factors, what is the reliability of this model for future prediction?

Response: The major weakness of the hydrodynamic model application is that item properties, such as size, shape and density are not in detail taken into account. We are work on improvements. The present simulations give an idea of what is happening, the dynamic and controlling factors, especially when having a look at the simulations videos (that shall be attached to this paper). However, its reliability is limited and cannot be assessed in detail.

We added the paragraph: “Supplementary Materials: The following supporting information can be downloaded at: https://doi.org/10.5281/zenodo.8433800, Video S1: Particle tracking simulations of near-bottom marine macro-litter transport in Rostock (Warnow Estuary).”

R1 comment: Please emphasis the research importance of such an elaborate survey.

Response: In the conclusion we added: “We provide the first comparable macro-litter pollution data for marinas and city harbor sea floors. The data is restricted to identifiable items located at the sediment surface.”. Further, we added in the introduction: “In Alexandria, Egypt, Shabaka et al. [22] describe high microplastic concentrations at beaches and floating near the water surface in harbors. In Tunisia, about 1 kg of waste is produced per inhabitant and day and it is assumed that a tourist generates more than twice as much [23]. In all North African countries high plastic and litter emissions into coastal and marine systems result from waste mismanagement [24]. It can be expected that intensive tourism, together with waste mismanagement and local fisheries causes a strong litter pollution of harbor sea floors in North Africa. Kühn et al. [25] provide a review of the consequences of marine litter in the marine environment and on wildlife, such as entanglement and ingestion. Another problem is the toxicity of chemicals associated especially with plastic litter [26]. On the other hand, marine litter is known to serve as habitat and dispersal vector [27]. Last not least, litter is a nuisance that negatively affects the public perception and can have negative consequences on tourism [28].”

We carried out another English language revision. 

Reviewer 2 Report

The study by Schernewski et al. investigated macro-litter pollution at 14 harbors and marinas in North Africa and the western Baltic Sea, with a simple underwater camera system. The study is comprehensive and provides useful insight into the pollution status in the studied areas. Below are some inquires/suggestions for the authors to consider.

1.     What are the major factors affecting the dispersion of macro-litters on the seafloor? Would the distance to the seashore and the depth of the seafloor be important factors? If possible, please provide some statistical analyses to elucidate these factors.

2.     How deep could the simple underwater camera system be used? In Table 1, please indicate the range of seafloor depth of the studied area.

Author Response

Thank you for your readiness to review our paper and for your constructive comments. Please find our response to your comments below:

R2 comment: What are the major factors affecting the dispersion of macro-litters on the seafloor? Would the distance to the seashore and the depth of the seafloor be important factors? If possible, please provide some statistical analyses to elucidate these factors.

Response: The major weakness of the hydrodynamic model application is that item properties, such as size, shape and density are not in detail taken into account. Therefore, the reliability of the simulations is limited, cannot be assessed in detail and statistics would not provide reliable insights. We are still working on adequate model improvements. However, the present simulations give an idea of what is happening, the dynamic and controlling factors. This is especially true when having a look at the simulations videos. Therefore, we attached the videos to this paper:

 “Supplementary Materials: The following supporting information can be downloaded at: https://doi.org/10.5281/zenodo.8433800, Video S1: Particle tracking simulations of near-bottom marine macro-litter transport in Rostock (Warnow Estuary).”

R2 comment: How deep could the simple underwater camera system be used? In Table 1, please indicate the range of seafloor depth of the studied area.

To avoid misunderstandings, we added a paragraph in the methods: “Before the sea floor pictures were taken, the suitable height of the camera above the sea bottom was determined. The acceptable distance between camera and sea floor were between 0.25 m and 1.6 m and depended on water transparency (Secchi depth) and water depth. In average the distance was about 40 % of the Secchi depth and about 35 % of the water depth. The present system was suitable for a maximum water depth up to 4 m, which was sufficient for all harbors.”

Additionally, Table 1 was extended and now shows the requested complementary data, namely Secchi depth (water transparency), water depth and the distance between camera lens and sea floor.

To better emphasis background of the research (especially in Africa) and its relevance we added in the conclusion: “We provide the first comparable macro-litter pollution data for marinas and city harbor sea floors. The data is restricted to identifiable items located at the sediment surface.”. Further, we added in the introduction: “In Alexandria, Egypt, Shabaka et al. [22] describe high microplastic concentrations at beaches and floating near the water surface in harbors. In Tunisia, about 1 kg of waste is produced per inhabitant and day and it is assumed that a tourist generates more than twice as much [23]. In all North African countries high plastic and litter emissions into coastal and marine systems result from waste mismanagement [24]. It can be expected that intensive tourism, together with waste mismanagement and local fisheries causes a strong litter pollution of harbor sea floors in North Africa. Kühn et al. [25] provide a review of the consequences of marine litter in the marine environment and on wildlife, such as entanglement and ingestion. Another problem is the toxicity of chemicals associated especially with plastic litter [26]. On the other hand, marine litter is known to serve as habitat and dispersal vector [27]. Last not least, litter is a nuisance that negatively affects the public perception and can have negative consequences on tourism [28].”

Reviewer 3 Report

This work provided a simple mobile underwater camera system for harbor sea floor pollution monitoring, and then explored the impact of several different environments on harbor sea floor pollution. The method seems convenient and effective, but there are some questions confusing me.

1.       Can this device detect wastes that hidden by sediment/sand? If not, does whether the waste is buried have a different impact on the environment?

2.       What are the dangers of macro-litter (plastic) sea bottom pollution? It is recommended that the author explain it at the beginning of the introduction section.

3.       The author emphasizes “plastic” in the article, but what I see from Figure 3 should be more glass or metal? So what is the result of the final statistical wastes?

4.       I'm curious how long it will take for waste to return to its current density after seafloor cleaning is carried out? Will a single storm make cleaning works wasted? I think this will be a topic of more interest to everyone.

None

Author Response

Thank you for your readiness to review our paper and for your constructive comments. Please find our response to your comments below:

We cannot improve the research design, but what we did is to be more explicit with respect to the methodology. Therefore, we added a paragraph:

“Before the sea floor pictures were taken, the suitable height of the camera above the sea bottom was determined. The acceptable distance between camera and sea floor were between 0.25 m and 1.6 m and depended on water transparency (Secchi depth) and water depth. In average the distance was about 40 % of the Secchi depth and about 35 % of the water depth. The present system was suitable for a maximum water depth up to 4 m, which was sufficient for all harbors.”

Additionally, Table 1 was extended and now shows complementary data, namely Secchi depth (water transparency), water depth and the distance between camera lens and sea floor.

R3 comment: Can this device detect wastes that hidden by sediment/sand? If not, does whether the waste is buried have a different impact on the environment?

Response: No, this is not possible. To clarify that we added in the conclusion: “We provide the first comparable macro-litter pollution data for marinas and city harbor sea floors. The data is restricted to identifiable items located at the sediment surface.”.

R3 comment: What are the dangers of macro-litter (plastic) sea bottom pollution? It is recommended that the author explain it at the beginning of the introduction section.

Response: To address this comment we added in the introduction: “In Alexandria, Egypt, Shabaka et al. [22] describe high microplastic concentrations at beaches and floating near the water surface in harbors. In Tunisia, about 1 kg of waste is produced per inhabitant and day and it is assumed that a tourist generates more than twice as much [23]. In all North African countries high plastic and litter emissions into coastal and marine systems result from waste mismanagement [24]. It can be expected that intensive tourism, together with waste mismanagement and local fisheries causes a strong litter pollution of harbor sea floors in North Africa. Kühn et al. [25] provide a review of the consequences of marine litter in the marine environment and on wildlife, such as entanglement and ingestion. Another problem is the toxicity of chemicals associated especially with plastic litter [26]. On the other hand, marine litter is known to serve as habitat and dispersal vector [27]. Last not least, litter is a nuisance that negatively affects the public perception and can have negative consequences on tourism [28].”

R3 comment: The author emphasizes “plastic” in the article, but what I see from Figure 3 should be more glass or metal? So what is the result of the final statistical wastes?

Response: The composition of items is shown in Table 2 “Item distribution of the marina bottom marine litter screening”. The photos in Figure 3 were chosen to indicate the variety of items.

R3 comment: I'm curious how long it will take for waste to return to its current density after seafloor cleaning is carried out? Will a single storm make cleaning works wasted? I think this will be a topic of more interest to everyone.

Response: This is something we cannot answer yet. The major weakness of the hydrodynamic model application is that item properties, such as size, shape and density are not in detail taken into account. Therefore, the reliability of the simulations is limited, cannot be assessed in detail. However, the present simulations give an idea of what is happening, the dynamic and controlling factors. This is especially true when having a look at the simulations videos. Therefore, we attached the videos to this paper.

“Supplementary Materials: The following supporting information can be downloaded at: https://doi.org/10.5281/zenodo.8433800, Video S1: Particle tracking simulations of near-bottom marine macro-litter transport in Rostock (Warnow Estuary).”

We are still working on adequate model improvements. Only a new model version and complementary sampling would allow to estimate how fast a pollution takes place and to what extent storms cause sediment surface cleanings.

We carried out another English language revision.

Round 2

Reviewer 3 Report

The manuscript has been sufficiently improved.